# Progression-free survival and overall survival after *BRCA1/2*-associated epithelial ovarian cancer: A matched cohort study

**Bernadette A. M. Heemskerk-Gerritsen**[1]*, **Antoinette Hollestelle**[1], **Christi J. van Asperen**[2], **Irma van den Beek**[3], **Willemien J. van Driel**[4], **Klaartje van Engelen**[5], **Encarna B. Gómez Garcia**[6], **Joanne A. de Hullu**[7], **Marco J. Koudijs**[8], **Marian J. E. Mourits**[9], **Maartje J. Hooning**[1◉], **Ingrid A. Boere**[1◉]

1 Department of Medical Oncology, Erasmus MC Cancer Institute, Rotterdam, the Netherlands, 2 Department of Clinical Genetics, Leiden University Medical Center, Leiden, the Netherlands, 3 Department of Human Genetics, Amsterdam University Medical Center (University of Amsterdam), Amsterdam, the Netherlands, 4 Department of Gynecology, Netherlands Cancer Institute, Amsterdam, the Netherlands, 5 Department of Clinical Genetics, Amsterdam University Medical Center (VUmc), Amsterdam, the Netherlands, 6 Department of Clinical Genetics, Maastricht University Medical Center, Maastricht, the Netherlands, 7 Department of Obstetrics & Gynecology, Radboud University Medical Center, Nijmegen, the Netherlands, 8 Department of Biomedical Genetics, Utrecht University Medical Center, Utrecht, the Netherlands, 9 Department of Gynecologic Oncology, University Medical Center Groningen, Groningen, the Netherlands

◉ These authors contributed equally to this work.
* b.heemskerk-gerritsen@erasmusmc.nl

**Data Availability Statement:** All relevant aggregated data are available within the manuscript

## Abstract

### Introduction

Germline *BRCA1/2*-associated epithelial ovarian cancer has been associated with better progression-free survival and overall survival than sporadic epithelial ovarian cancer, but conclusive data are lacking.

### Methods

We matched 389 *BRCA1*-associated and 123 *BRCA2*-associated epithelial ovarian cancer patients 1:1 to sporadic epithelial ovarian cancer patients on year of birth, year of diagnosis, and FIGO stage (< = IIA/> = IIB). Germline DNA test was performed before or after epithelial ovarian cancer diagnosis. All patients received chemotherapy. We used Cox proportional hazards models to estimate the associations between mutation status (*BRCA1* or *BRCA2* versus sporadic) and progression-free survival and overall survival. To investigate whether DNA testing after epithelial ovarian cancer diagnosis resulted in survival bias, we performed additional analyses limited to *BRCA1/2*-associated epithelial ovarian cancer patients with a DNA test result before cancer diagnosis (n = 73 *BRCA1*; n = 9 *BRCA2*) and their matched sporadic controls.

### Results

The median follow-up was 4.4 years (range 0.1–30.1). During the first three years after epithelial ovarian cancer diagnosis, progression-free survival was better for *BRCA1* (HR 0.88,

and its Supporting information files. We have
ethical restrictions about openly releasing the data
set to the public as the data contain potentially
identifying or sensitive patient information. The
ethical restrictions were imposed by the individual
participating Dutch academic centers, the
Netherlands Cancer Institute, and the Netherlands
Comprehensive Cancer Organization (IKNL).
However, to facilitate the replication of results, the
anonymized data are available upon reasonable
request. Requests for the complete data can be
made to the Hereditary Breast and Ovarian Cancer
Research Group Netherlands (HEBON) at the
Netherlands Cancer Institute (hebon.resource@nki.
nl).

**Funding:** This work was supported by a Grant from
the Dutch Cancer Society (Grant No. EMCR2014-
6699). The HEBON study is supported by the
Dutch Cancer Society (Grant Nos. NKI1998-1854,
NKI2004-3088, NKI2007-3756), the Netherlands
Organisation of Scientific Research (Grant No.
NWO 91109024), the Dutch Pink Ribbon
foundation (Grant Nos. 110005 and 2014-187.
WO76), BBMRI (Grant No. NWO 184.021.007/
CP46) and Transcan (Grant No. JTC 2012 Cancer
12-054). The funders had no role in study design,
data collection and analysis, decision to publish, or
preparation of the manuscript.

**Competing interests:** The authors have declared
that no competing interests exist.

95% CI 0.74–1.04) and *BRCA2* (HR 0.58, 95% CI 0.41–0.81) patients than for sporadic patients. Overall survival was better during the first six years after epithelial ovarian cancer for *BRCA1* (HR 0.7, 95% CI 0.58–0.84) and *BRCA2* (HR 0.41, 95% CI 0.29–0.59) patients. After surviving these years, survival benefits disappeared or were in favor of the sporadic patients.

## Conclusion

For epithelial ovarian cancer patients who received chemotherapy, we confirmed survival benefit for *BRCA1* and *BRCA2* germline pathogenic variant carriers. This may indicate higher sensitivity to chemotherapy, both in first line treatment and in the recurrent setting. The observed benefit appears to be limited to a relatively short period after epithelial ovarian cancer diagnosis.

## Introduction

Despite a relatively low cumulative life-time risk–~1.6% for women in the western world–ovarian cancer is the fifth most common cause of cancer death in women, with worldwide over 150,000 deaths each year [1, 2]. The high mortality rate is largely due to the tendency to early spreading in the abdominal cavity, and most ovarian cancers being diagnosed at advanced stages (FIGO stage III/IV) [3–5]. Despite a high response rate to platinum-based chemotherapy, the overall survival (OS) remains poor with a 5-year overall survival of only 30–40% [3, 4].

Approximately 11–15% of all epithelial ovarian cancer (EOC) patients carry a *BRCA1* or *BRCA2* germline pathogenic variant (gPV) [6–10]. Women with a *BRCA* gPV have a high cumulative life-time risk of developing EOC, being 40–60% for *BRCA1* and 10–25% for *BRCA2* gPV carriers [11–15]. In general, EOC in *BRCA* gPV carriers is diagnosed at a younger age than in sporadic patients, and younger in *BRCA1* gPV carriers than in *BRCA2* gPV carriers [11–16]. In view of the absence of effective screening for EOC, women with a proven *BRCA* gPV are advised to opt for premenopausal risk-reducing salpingo-oophorectomy at the age of 35 to 40 years for *BRCA1* gPV carriers and 40 to 45 years for *BRCA2* gPV carriers.

*BRCA*-deficiency is associated with an impaired ability to repair double-strand DNA breaks by the DNA repair mechanism homologous recombination [17–21]. This may lead to higher response rates to first-line platinum-based chemotherapy–which causes double-strand DNA breaks–and thus to improved survival [22–24]. Indeed, some studies have reported better survival for *BRCA*-associated EOC patients than for sporadic patients [10, 22–26], although the reported results are not consistent [27–29]. Survival benefit may be limited to *BRCA2* gPV carriers [30], or only applicable to the first five to ten years [31–33]. A few studies showed also higher response rates to platinum-based chemotherapy after recurrent EOC in *BRCA* gPV carriers than in sporadic EOC patients, but the numbers of included patients are small [10, 24, 34]. Further, the sensitivity to platinum-based chemotherapy might depend on the associated gene and/or the specific pathogenic variant [28, 30].

Altogether, definitive evidence of better prognosis for *BRCA*-associated EOC patients is still unavailable. Moreover, while *BRCA1* and *BRCA2* tumors might represent different entities, most studies did not investigate prognosis and survival after EOC separately for *BRCA1* and *BRCA2*. In the current retrospective cohort study we compare progression-free survival (PFS)

and overall survival (OS) between either germline *BRCA1*-associated EOC patients or germline *BRCA2*-associated EOC patients and matched sporadic EOC patients treated with first-line chemotherapy.

## Participants and methods

### Study population

For this retrospective matched cohort study, we selected *BRCA1* and *BRCA2* gPV carriers with a history of EOC from the national Hereditary Breast and Ovarian Cancer Netherlands (HEBON) database. In the context of the HEBON study, members of breast and/or ovarian cancer families are being identified through the departments of Clinical Genetics/Family Cancer Clinics at eight Dutch academic centers and the Netherlands Cancer Institute [35]. The study was approved by the Medical Ethical Committees of all participating centers. Written informed consent was obtained from each participant or from a close relative in case of deceased individuals. Relevant data on participants including data on preventive strategies, the occurrence of cancer, and vital status were retrieved and updated through medical files and questionnaires, and through linkages to the Netherlands Cancer Registry, the Dutch Pathology Database, and the municipal registry database. The latest follow-up date was December 31, 2017.

From this national cohort, we selected 389 *BRCA1* gPV carriers and 123 *BRCA2* gPV carriers with EOC. Patients were eligible for the study if they were diagnosed with EOC after 1988, had a proven *BRCA* gPV (with DNA test result either before or after EOC diagnosis), and received chemotherapy after diagnosis of primary EOC (in case of surgery, either before or after).

The selected *BRCA* gPV carriers were matched 1:1 to sporadic EOC patients from the National Cancer Registry on year of birth (+/− 5 years), year of EOC diagnosis (+/- 5 years), and FIGO stage (≤IIA/≥IIB/unknown). Sporadic patients were defined as patients who were either not DNA tested due to a negative family history of breast cancer or ovarian cancer or because DNA testing was not available yet, or DNA tested and without a proven *BRCA* gPV. Notably, about 5% of EOCs have a somatic *BRCA* pathogenic variant, but somatic testing has only been widely implied since 2020 in the Netherlands and data hereon is therefore not available for the current cohort.

### Data collection

We retrieved data on the associated gene (i.e. *BRCA1* or *BRCA2*) and date of DNA test result, dates of birth and death, and dates of diagnosis of EOC, first recurrent disease, and other cancers. We also collected data on tumor characteristics (FIGO stage, histology, and differentiation grade), CA125 at EOC diagnosis, and treatment details after primary EOC diagnosis and in the recurrent setting (surgery, type of chemotherapy, and maintenance treatment with poly (ADP-ribose) polymerase inhibitors (PARPi)).

### Statistical analyses

We evaluated clinical characteristics by comparing EOC patients with (*BRCA1* and *BRCA2* groups) and without a proven *BRCA* gPV (sporadic groups). We used Pearson's chi-squared test for differences between the *BRCA* groups and the sporadic groups for categorical variables, and Wilcoxon rank-sum to test the equality of the medians for continuous variables.

The outcomes PFS and OS were measured in person-years of observation. The observation period started at the date of EOC diagnosis, and ended at the date of a censoring event or the

date of first recurrence for the PFS analyses or death for the OS analyses. Censoring events included diagnosis of a new primary malignant tumor, date of last follow-up, and date of death (for PFS only).

To estimate the associations between gPV status (*BRCA1* or *BRCA2* versus sporadic) and survival endpoints, we used Cox proportional hazards models with the sporadic groups as the references to obtain hazard ratios (HR) with corresponding 95% confidence intervals (CI). We considered age at EOC diagnosis, grade, CA125 at diagnosis, type of chemotherapy, debulking surgery (yes/no), and complete debulking surgery (yes/no, i.e. the absence/presence of any residual disease) as potential confounders. The matching variables year of birth, year of EOC diagnosis, and FIGO stage were by definition not confounding factors. We generated Kaplan-Meier survival curves, and used the log-rank test for equality of survivor functions to test whether the curves were significantly different from each other. We performed all analyses separately for *BRCA1* and *BRCA2* gPV carriers.

Further, *BRCA* gPV carriers who underwent DNA testing after EOC diagnosis survived at least until this DNA test, which was in some cases many years after EOC diagnosis. To investigate whether this resulted in survival bias in favor of the *BRCA* gPV carriers, we also performed prospective analyses limited to *BRCA*-associated EOC patients with a DNA test result before EOC diagnosis and their matched sporadic controls.

Furthermore, as previous studies reported different short-term and long-term survival rates for gPV carriers [31–33], the proportional hazards assumption may be violated. Therefore we used Schoenfeld residuals to test whether the proportional hazards assumption is violated. If that was the case, we stratified the Cox models by a specified time-of-observation, i.e. the moment $t$ where the HR switched from under 1 to above 1 (or vice versa). We calculated this exact moment using the formula

$$HR(t) = exp(\beta x + \delta x t) \tag{1}$$

where $x$ is the variable of interest (i.e. *BRCA1*/*BRCA2* or sporadic), β is the β coefficient, and δ is the time-varying coefficient. When the proportional hazards assumption is valid, δ equals zero. Otherwise, we can calculate $t$ using

$HR(t) = 1$

$\Rightarrow ln(HR(t)) = 0$

$\Rightarrow ln(exp(\beta x + \delta x t)) = 0$

$\Rightarrow \beta x + \delta x t = 0$

$\Rightarrow t = -\beta / \delta$

where β and δ are derived from the Cox model including both the variable for gPV status and the interaction term of this variable with time.

All p-values were two-sided, and a significance level α = 0.05 was used. Analyses were performed using Stata/SE (version 16.0, StataCorp, Collegestation, TX).

## Results

### Study population

As shown in Table 1, the 389 *BRCA1* gPV carriers and the 123 *BRCA2* gPV carriers had longer follow-up than their matched sporadic EOC patients (median years 4.8 versus 3.5 for the *BRCA1* comparison, p<0.001; 5.7 versus 3.5 for the *BRCA2* comparison, p<0.001). The vast majority of the patients received platinum-based chemotherapy. After recurrence of disease,

**Table 1. Patient and tumor characteristics.**

| | BRCA1 | | Sporadic | | | BRCA2 | | Sporadic | | |
|---|---|---|---|---|---|---|---|---|---|---|
| | N | % | N | % | p-value | N | % | N | % | p-value |
| | 389 | | 389 | | | 123 | | 123 | | |
| **Follow-up, median years (range)** | 4.8 | (0.1–26.7) | 3.5 | 0.1–30.1 | <0.001 | 5.7 | (0.5–25.6) | 3.5 | (0.1–24.1) | <0.001 |
| **Year of birth, median (range)** | 1950 | (1922–1981) | 1950 | (1922–1979) | .757 | 1946 | (1923–1972) | 1946 | (1922–1972) | .824 |
| **DNA test result** | | | | | | | | | | |
| Median age, median (range) | 54 | (26–81) | | | | 61 | (35–79) | | | |
| Time between EOC diagnosis and DNA test result, median years (range) | 1 | (0–19.8) | | | | 1.1 | (0–16.3) | | | |
| before EOC | 72 | (19%) | | | | 8 | (7%) | | | |
| <6 months | 46 | (12%) | | | | 19 | (15%) | | | |
| 6–12 months | 74 | (19%) | | | | 24 | (20%) | | | |
| 1–3 years | 117 | (30%) | | | | 38 | (31%) | | | |
| 3–5 years | 28 | (7%) | | | | 15 | (12%) | | | |
| 5–10 years | 34 | (9%) | | | | 11 | (9%) | | | |
| >10 years | 17 | (4%) | | | | 7 | (6%) | | | |
| unknown | 1 | (0%) | | | | 1 | (1%) | | | |
| **Year of EOC diagnosis, median (range)** | 2004 | (1989–2015) | 2004 | (1989–2014) | .726 | 2004 | (1989–2014) | 2005 | (1989–2014) | .558 |
| **Age at EOC diagnosis, median (range)** | 52 | (23–78) | 52 | (23–77) | .488 | 58 | (35–76) | 57 | (35–79) | .888 |
| **FIGO** | | | | | | | | | | |
| Low (≤IIA) | 34 | (10%) | 46 | (13%) | .151 | 16 | (14%) | 19 | (17%) | .522 |
| High (≥IIB) | 323 | (90%) | 310 | (87%) | | 96 | (86%) | 90 | (83%) | |
| unknown | 32 | | 33 | | | 11 | | 14 | | |
| **Grade** | | | | | | | | | | |
| Well differentiated | 9 | (3%) | 32 | (11%) | <0.05 | 1 | (1%) | 5 | (5%) | .088 |
| Poorly differentiated | 320 | (97%) | 261 | (89%) | | 102 | (99%) | 94 | (95%) | |
| unknown | 60 | | 96 | | | 14 | | 24 | | |
| **Histology** | | | | | | | | | | |
| Serous | 282 | (73%) | 224 | (58%) | <0.001 | 81 | (67%) | 81 | (66%) | .546 |
| Endometrioid | 30 | (8%) | 53 | (14%) | | 8 | (7%) | 15 | (12%) | |
| Clear cell | 3 | (1%) | 23 | (6%) | | 3 | (2%) | 3 | (2%) | |
| Mucinous | 7 | (2%) | 18 | (4%) | | 3 | (2%) | 5 | (4%) | |
| Adenocarcinoma NOS | 52 | (13%) | 61 | (16%) | | 24 | (20%) | 17 | (14%) | |
| Other | 11 | (3%) | 8 | (2%) | | 3 | (2%) | 2 | (2%) | |
| Unknown | 4 | | 2 | | | 1 | | 0 | | |
| **CA125 (U/ml)** | | | | | | | | | | |
| ≤35 | 33 | (12%) | 19 | (6%) | <0.01 | 4 | (5%) | 9 | (9%) | .576 |
| 35–500 | 106 | (37%) | 148 | (49%) | | 33 | (41%) | 38 | (37%) | |
| >500 | 147 | (51%) | 137 | (45%) | | 44 | (54%) | 55 | (54%) | |
| unknown | 103 | | 85 | | | 42 | | 21 | | |
| **Type of chemotherapy** | | | | | | | | | | |
| platinum & anthracyclines | 1 | (0%) | 2 | (1%) | .527 | 0 | (0%) | 1 | (1%) | .521 |
| platinum & taxanen | 313 | (84%) | 301 | (83%) | | 102 | (87%) | 96 | (81%) | |
| platinum | 52 | (14%) | 49 | (13%) | | 14 | (12%) | 20 | (17%) | |
| taxanen & anthracyclines | 1 | (0%) | 5 | (1%) | | 0 | (0%) | 1 | (1%) | |
| taxanen | 6 | (2%) | 6 | (2%) | | 1 | (1%) | 1 | (1%) | |
| unknown | 16 | | 26 | | | 6 | | 4 | | |

(*Continued*)

**Table 1.** (*Continued*)

| | *BRCA1* | | Sporadic | | | *BRCA2* | | Sporadic | | |
|---|---|---|---|---|---|---|---|---|---|---|
| | N | % | N | % | p-value | N | % | N | % | p-value |
| | **389** | | **389** | | | **123** | | **123** | | |
| **Timing of chemotherapy** | | | | | | | | | | |
| Neoadjuvant | 64 | (18%) | 83 | (24%) | .051 | 35 | (30%) | 27 | (24%) | .304 |
| Adjuvant | 299 | (82%) | 270 | (76%) | | 82 | (70%) | 86 | (76%) | |
| unknown | 26 | | 36 | | | 6 | | 10 | | |
| **Debulking surgery** | | | | | | | | | | |
| No | 10 | (3%) | 26 | (7%) | <0.01 | 3 | (2%) | 7 | (6%) | .201 |
| Yes (primary or interval) | 378 | (97%) | 354 | (93%) | | 118 | (98%) | 115 | (94%) | |
| unknown | 1 | | 9 | | | 2 | | 1 | | |
| **Complete debulking** | | | | | | | | | | |
| No | 127 | (46%) | 110 | (47%) | .893 | 38 | (48%) | 34 | (47%) | .851 |
| Yes | 149 | (54%) | 126 | (53%) | | 41 | (52%) | 39 | (53%) | |
| unknown | 102 | | 118 | | | 39 | | 42 | | |
| **Recurrent disease** | 299 | (77%) | 306 | (79%) | .546 | 84 | (68%) | 92 | (75%) | .258 |
| Age at 1st recurrence, median (range) | 54 | (29–79) | 55 | (30–78) | .184 | 61 | (35–79) | 60 | (37–79) | .817 |
| Year of 1st recurrence, median (range) | 2006 | (1990–2017) | 2006 | (1989–2020) | .276 | 2007 | (1994–2014) | 2007 | (1989–2019) | .828 |
| Time between diagnosis of EOC and 1st recurrence, median months (range) | 18.3 | (0.6–179.3) | 15.9 | (0.5–364.3) | <0.005 | 22.3 | (2.1–116.7) | 15.7 | (0.6–174.1) | <0.001 |
| Before DNA test result | 108 | (36%) | | | | 31 | (37%) | | | |
| After DNA test result | 191 | (64%) | | | | 52 | (63%) | | | |
| Chemotherapy after recurrence | | | | | | | | | | |
| No | 31 | (10%) | 56 | (21%) | <0.001 | 11 | (13%) | 11 | (14%) | .826 |
| Yes | 266 | (90%) | 206 | (79%) | | 73 | (87%) | 66 | (86%) | |
| Unknown | 2 | | 44 | | | 0 | | 15 | | |
| PARPi after recurrence | | | | | | | | | | |
| No | 269 | (91%) | 254 | (98%) | <0.001 | 75 | (91%) | 69 | (96%) | .272 |
| Yes | 25 | (9%) | 4 | (2%) | | 7 | (9%) | 3 | (4%) | |
| Unknown | 5 | | 48 | | | 2 | | 20 | | |
| **Deceased** | 274 | (70%) | 292 | (75%) | .147 | 78 | (63%) | 91 | (74%) | .074 |
| Age at death, median (range) | 57 | (32–83) | 56 | (33–87) | .248 | 63 | (36–89) | 62 | (38–82) | .121 |
| Time between 1st recurrence and death, median months (range) | 25.9 | (0–166) | 13.9 | (0–156.1) | <0.001 | 25 | (0.3–152.2) | 15.3 | (0–106.8) | <0.001 |
| Time between diagnosis of EOC and death, median months (range) | 49.2 | (0.6–233.9) | 33.4 | (0.9–217.8) | <0.001 | 53 | (9.3–254.7) | 32.9 | (0.6–277.6) | <0.001 |

Abbreviations: EOC, epithelial ovarian cancer; PARPi, poly(ADP-ribose) polymerase inhibitors.

*BRCA1*-associated EOC patients were treated more often with chemotherapy than the sporadic patients (90% versus 79%, p<0.001), which did not apply for the *BRCA2* comparison (Table 1). The characteristics for the dataset used for the prospective analyses (in total n = 82 matched pairs) are shown in S1 Table.

## Potential confounders

No differences between the groups were observed for the matching variables year of birth, year of EOC diagnosis, and FIGO stage, nor for age at EOC diagnosis and type of chemotherapy (Table 1). Due to the large proportion of missing data for CA125 at diagnosis, EOC grade, and

completeness of debulking surgery, no adjustment was possible for these variables. We performed Cox models adjusted for debulking surgery (yes/no), with the sporadic groups as the references.

## Survival analyses

We observed no differences between the *BRCA*-associated groups and their matched sporadic patients in the percentage of patients with recurrent disease. The time between diagnoses of EOC and first recurrence, though, was longer for *BRCA*-associated patients than for the sporadic patients (*BRCA1* comparison: 18.3 versus 15.9 months, p<0.005; *BRCA2* comparison: 22.3 versus 15.7 months, p<0.001; Table 1). Likewise, the percentages of deceased patients were similar in all comparison groups, while the time between diagnosis of first recurrence and death is longer for EOC patients with a *BRCA* gPV (*BRCA1* comparison: 25.9 versus 13.9 months, p<0.001; *BRCA2* comparison: 25 versus 15.3 months, p<0.001; Table 1).

As shown in Table 2, while *BRCA1* gPV status was not associated with significant differences in PFS, the Cox model for OS yielded an HR of 0.82 (95% CI 0.7–0.97) in favor of *BRCA1* gPV carriers. In addition, the prospective analyses–limited to *BRCA1*-associated EOC patients with a DNA test result before EOC diagnosis and their matched sporadic controls–showed better PFS (HR 0.65, 95% CI 0.43–0.97), but no significant OS benefit for *BRCA1* gPV carriers (HR 0.86, 95% CI 0.56–1.3; Table 2). Accompanying survival curves are depicted in

**Table 2. Association of *BRCA1* germline pathogenic variant status with progression-free survival and overall survival.**

| | Progression-free survival | | | | | Overall survival | | | | |
|---|---|---|---|---|---|---|---|---|---|---|
| | N | PYO | Events | Rec. rate[1] (95% CI) | HR (95% CI)[2] | N | PYO | Events | Mort. rate[1] (95% CI) | HR (95% CI)[2] |
| *Complete analyses* | | | | | | | | | | |
| **Total observation period** | | | | | | | | | | |
| *BRCA1* | 389 | 1475 | 299 | 203 (181–227) | 0.9 (0.77–1.06) | 389 | 2452 | 274 | 112 (99–126) | 0.82 (0.7–0.97) |
| sporadic | 389 | 1527 | 306 | 200 (179–224) | 1 | 389 | 2244 | 292 | 130 (116–146) | 1 |
| **Observation period < t** | t = 3.3 yrs | | | | | t = 6 yrs | | | | |
| *BRCA1* | 389 | 807 | 249 | 309 (273–349) | 0.88 (0.74–1.04) | 389 | 1702 | 206 | 121 (106–139) | 0.7 (0.58–0.84) |
| sporadic | 389 | 726 | 251 | 346 (305–391) | 1 | 389 | 1416 | 251 | 177 (157–201) | 1 |
| **Observation period ≥ t** | | | | | | | | | | |
| *BRCA1* | 124 | 668 | 50 | 75 (57–99) | 1.01 (0.69–1.49) | 150 | 751 | 68 | 91 (71–115) | 1.61 (1.09–2.38) |
| sporadic | 120 | 800 | 55 | 69 (53–90) | 1 | 120 | 828 | 41 | 50 (36–67) | 1 |
| *Prospective analyses* | | | | | | | | | | |
| **Total observation period** | | | | | | | | | | |
| *BRCA1* | 73 | 268 | 47 | 176 (132–234) | 0.65 (0.43–0.97) | 73 | 406 | 42 | 103 (76–140) | 0.86 (0.56–1.3) |
| sporadic | 73 | 252 | 55 | 218 (167–284) | 1 | 73 | 380 | 47 | 124 (93–164) | 1 |
| **Observation period < t** | t = 3.1 yrs | | | | | t = 5.7 yrs | | | | |
| *BRCA1* | 73 | 155 | 36 | 232 (168–322) | 0.56 (0.36–0.88) | 73 | 298 | 32 | 107 (76–152) | 0.72 (0.45–1.15) |
| sporadic | 73 | 131 | 46 | 351 (263–469) | 1 | 73 | 264 | 41 | 155 (114–210) | 1 |
| **Observation period ≥ t** | | | | | | | | | | |
| *BRCA1* | 31 | 113 | 11 | 97 (54–176) | 1.15 (0.47–2.78)[3] | 27 | 109 | 10 | 92 (50–171) | 1.89 (0.67–5.31)[3] |
| sporadic | 24 | 121 | 9 | 74 (39–176) | 1 | 25 | 116 | 6 | 52 (23–116) | 1 |

Abbreviations: N, number of patients; PYO, person-years of observation; Rec. rate, recurrence rate; Mort. Rate, mortality rate; HR, hazard ratio; 95% CI, 95% confidence interval; *t*, time point where HR switches from under to above 1 (in years of observation after diagnosis of epithelial ovarian cancer).

[1] per 1000 PYO.

[2] adjusted for debulking surgery (yes/no).

[3] univariable analysis; adjusting for debulking surgery omitted due to zero patients without debulking surgery.

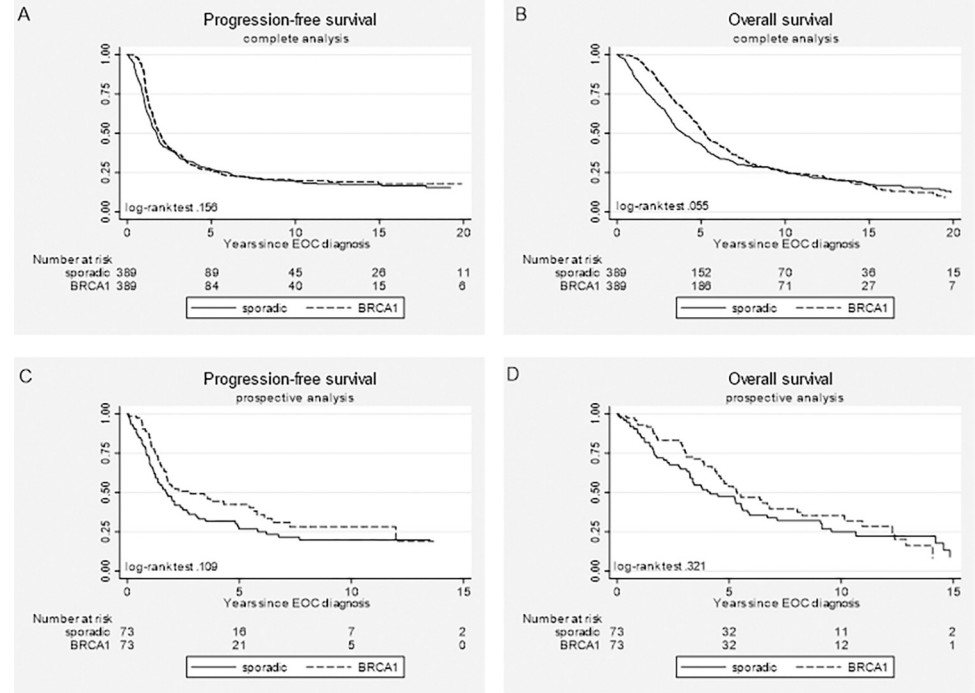

**Fig 1. Kaplan-Meier survival curves for *BRCA1*-associated epithelial ovarian cancer (EOC) patients (dashed lines) and sporadic EOC patients (solid lines) treated with chemotherapy.** (A) progression-free survival and (B) overall survival for the complete dataset; (C) progression-free survival and (D) overall survival for the prospective dataset.

Fig 1. As the proportional hazards assumption was violated for all models, the analyses were stratified for the moment in time *t* where the HR equals 1. The stratified analyses revealed HRs under 1 when the observation time was shorter than *t* (varying from 3.1 to 6 years), and above 1 for longer observation time (Table 2).

Overall, as shown in Table 3, *BRCA2* gPV carriers showed better PFS (HR 0.67, 95% CI 0.5–0.91) and OS (HR 0.61, 95% CI 0.44–0.83) than their matched sporadic patients, which can also be seen in Fig 2. The stratified analyses revealed a significant risk-reduction in favor of *BRCA2* gPV carriers for PFS (HR 0.58, 95% CI 0.41–0.81; Table 3) and OS (HR 0.41, 95% CI 0.29–0.59) for the observation period under *t* (being 3 and 6 years, respectively), but a higher risk for death after *t* (HR 3.14, 95% CI 1.34–7.34). The numbers of patients in the prospective analyses were too small to draw meaningful conclusions (Table 3 and Fig 2).

## Discussion

In this retrospective matched cohort study, we confirmed better PFS during the first three years after EOC diagnosis and OS benefit during the first six years for patients with a *BRCA1* or *BRCA2* germline PV. After surviving this period, the benefit disappears, and might even turn into a higher risk of dying for gPV carriers. The observed survival benefit was slightly stronger for *BRCA2* than for *BRCA1*.

Our results are in line with a number of previous studies. Studies with limited follow-up periods showed improved PFS and OS–with comparable periods without progression and time till death as seen in our study–for *BRCA1*-associated EOC patients [23], *BRCA2*-associated patients [23, 30], or combined *BRCA1/2* cohorts [10, 22, 24–26]. Studies with long-term periods of follow-up showed that improved overall survival seems to be mainly driven by the

**Table 3. Association of *BRCA2* germline pathogenic variant status with progression-free survival and overall survival.**

| | \multicolumn Progression-free survival | | | | | Overall survival | | | | |
|---|---|---|---|---|---|---|---|---|---|---|
| | N | PYO | Events | Rec. rate[1] (95% CI) | HR (95% CI)[2] | N | PYO | Events | Mort. rate[1] (95% CI) | HR (95% CI)[2] |
| *Complete analyses* | | | | | | | | | | |
| **Total observation period** | | | | | | | | | | |
| *BRCA2* | 123 | 538 | 84 | 156 (126–193) | 0.67 (0.5–0.91) | 123 | 882 | 78 | 88 (71–110) | 0.61 (0.44–0.83) |
| sporadic | 123 | 482 | 92 | 191 (156–234) | 1 | 123 | 702 | 91 | 130 (105–159) | 1 |
| **Observation period < *t*** | *t* = 3 yrs | | | | | *t* = 6 yrs | | | | |
| *BRCA2* | 123 | 269 | 65 | 242 (190–308) | 0.58 (0.41–0.81) | 123 | 570 | 52 | 91 (70–120) | 0.41 (0.29–0.59) |
| sporadic | 123 | 220 | 80 | 364 (292–453) | 1 | 123 | 450 | 84 | 186 (151–231) | 1 |
| **Observation period ≥ *t*** | | | | | | | | | | |
| *BRCA2* | 52 | 269 | 19 | 71 (45–111) | 1.37 (0.66–2.83)[3] | 57 | 312 | 26 | 83 (57–122) | 3.14 (1.34–7.34)[3] |
| sporadic | 38 | 262 | 12 | 45 (26–81) | 1 | 34 | 252 | 7 | 28 (13–58) | 1 |
| *Prospective analyses* | | | | | | | | | | |
| **Total observation period[4]** | | | | | | | | | | |
| *BRCA2* | 9 | 31 | 5 | 160 (66–383) | 0.64 (0.19–2.17)[3] | 9 | 42 | 5 | 118 (49–283) | 0.71 (0.2–2.54)[3] |
| sporadic | 9 | 22 | 6 | 278 (125–618) | 1 | 9 | 38 | 6 | 156 (70–347) | 1 |

Abbreviations: N, number of patients; PYO, person-years of observation; Rec. rate, recurrence rate; Mort. Rate, mortality rate; HR, hazard ratio; 95% CI, 95% confidence interval; *t*, time point where HR switches from under to above 1 (in years of observation after diagnosis of epithelial ovarian cancer).

[1] per 1000 PYO.

[2] adjusted for debulking surgery (yes/no).

[3] univariable analysis; adjusting for debulking surgery omitted due to zero patients without debulking surgery.

[4] for the prospective analyses, the proportional hazard assumption is satisfied: no stratified Cox model necessary.

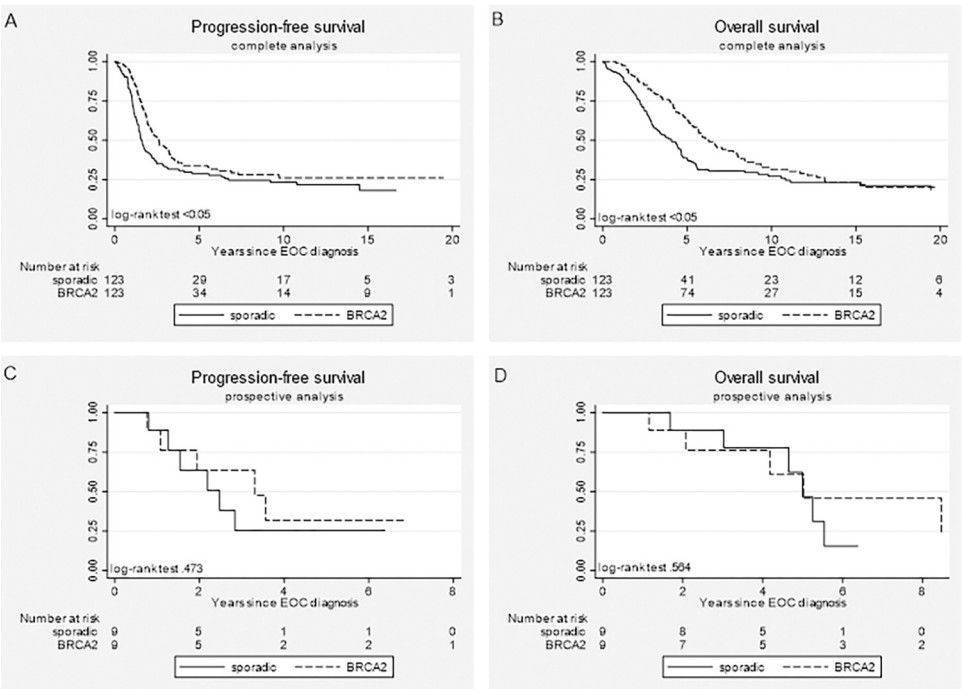

**Fig 2. Kaplan-Meier survival curves for *BRCA2*-associated epithelial ovarian cancer (EOC) patients (dashed lines) and sporadic EOC patients (solid lines) treated with chemotherapy.** (A) progression-free survival and (B) overall survival for the complete dataset; (C) progression-free survival and (D) overall survival for the prospective dataset.

first five years after diagnosis, with no benefit for those surviving that first period [31, 33, 36], or even worse OS afterwards [32], as observed in the current study.

Previously observed survival benefit could have been an age-effect. Recently, Mallen et al. observed worse survival for older patients, although the authors noted this may merely be the result of tumor biology rather than comorbidities [37]. As we currently matched–indirectly by matching on year of birth and year of diagnosis–on age at diagnosis, in contrast to most of the previous studies, our results support the suggestion that the observed difference is not related to age.

Our results support the hypothesis regarding *BRCA1/2*-associated EOC patients being more sensitive to platinum-based chemotherapy, especially since none of the patients in the current cohort received first-line maintenance treatment with a PARPi. Primary systemic treatment was not different for EOC patients with and without a *BRCA1/2* germline gPV. Therefore, differences in PFS cannot be attributed to differences in received chemotherapy treatment, leaving gPV status as the most likely explanation. Although debulking surgery was performed in the vast majority of the patients (~95%), sporadic patients underwent less often debulking surgery, possibly due to a very poor prognosis of disease at diagnosis, or due to the presence of comorbidities. As this may indicate a higher baseline risk for death in the sporadic EOC groups, we adjusted the analyses for this variable.

In the recurrent setting *BRCA*-associated EOC patients were more often treated with che-motherapy, which may have influenced survival. The rationale for omitting systemic treatment may have been a worse clinical situation at presentation of recurrent disease, potentially result-ing in a higher baseline risk of dying after recurrent EOC in the sporadic group. Further, in the *BRCA* groups more patients received PARPi after recurrent disease as a maintenance ther-apy. However, since PARPi has only been administered since 2015, the majority of the patients in the current cohort (approximately 95%) did not receive PARPi. For the sake of complete-ness, we performed subgroup analyses for OS excluding patients who were treated with PARPi in the recurrent setting and their matched counterparts. As none of the patients were treated with PARPi in the first-line, such subgroup analyses were not necessary for PFS. The subgroup analyses for OS revealed similar results as the original analyses (HR 0.81, 95% CI 0.68–0.97 for *BRCA1* and HR 0.65, 95% CI 0.47–0.9 for *BRCA2*). Thus, the influence of PARPi in the recur-rent setting is very limited in this study. This will increase, though, in future studies due to cur-rent clinical practice [38].

As previously described, including prevalent cases in studies involving *BRCA* gPV carriers can introduce seriously biased results [39, 40]. The majority of the gPV carriers in our cohort had their DNA tested after EOC diagnosis, with survival times up to 20 years until DNA test. Reassuringly, for the *BRCA1* comparison, the additional prospective analyses showed compa-rable overall survival as for the complete analyses, suggesting minimal bias as a result from delayed DNA testing. Unfortunately, due to the small number of *BRCA2* gPV carriers with a DNA test before cancer diagnosis, we were unable to draw meaningful conclusions from the prospective analyses among *BRCA2* gPV carriers and matched sporadic patients. Alternatively, we performed left-truncated analyses with the observation for the *BRCA* groups starting at the date of either DNA test result or EOC diagnosis, whichever came last, thus excluding patients with recurrent disease, LFU or death before DNA test result. As shown in S2 Table, the results were comparable to those for the complete and prospective analyses, confirming minimal bias due to delayed DNA testing.

Other strengths of the current design include the separate *BRCA1* and *BRCA2* analyses, and the fact that we matched–indirectly by matching on year of birth and year of diagnosis–on age at diagnosis. The advantage of the latter is that also the sporadic EOC patients were relatively young at diagnosis. Therefore, we can reasonably assume that the leading cause of death is

ovarian cancer in both groups, and that the mortality in the sporadic group is not distorted by competing causes of death due to older age.

One of the limitations of the study may be that not all sporadic patients were tested for a *BRCA* germline gPV. As mentioned before, the majority of the *BRCA*-associated EOC patients were tested after cancer diagnosis. Theoretically, the sporadic group may contain patients who were actually gPV carriers but never had the opportunity to get tested because they had died before DNA testing was performed or was even implemented in clinical practice. Unintentional misclassification of deceased patients in the sporadic groups may simultaneously overestimate the risk of dying in the sporadic groups and underestimate that risk in the *BRCA* groups, thus potentially overestimating the benefit for *BRCA* gPV carriers. Oppositely, the sporadic group may also include untested gPV carriers without recurrent disease or death, oppositely leading to an underestimation of the benefit. With regard to potential misclassification, we would like to emphasize that with the introduction of PARPi in 2015, more and more EOC patients undergo DNA testing sooner after diagnosis in order to receive the optimal treatment, at first only in the recurrent setting but nowadays also at primary disease.

In addition, data on somatic testing is not available for the current cohort. As a result, the sporadic group may contain a number of *BRCA* positive specimens, which may have influenced the results. However, as about only 5% of EOCs have a somatic *BRCA* pathogenic variant, we think this influence will be limited. Moreover, under the assumption that survival benefit will also apply to EOCs with a somatic *BRCA* pathogenic variant, potential misclassification of these EOCs in the sporadic group would led to an underestimation of the observed survival benefit on the short-term rather than an overestimation. Therefore, although the lack of data on somatic testing may be a deficiency in the study, in our opinion this may play a minor role.

Another limitation is the limited availability of data regarding complete debulking or residual disease after primary surgery in our study. In a previous study the only independent prognostic factor for survival in *BRCA1/2* gPV carriers was the extent of debulking at primary surgery, with better survival for patients without macroscopic disease [10]. Recently, Ataseven et al. confirmed that complete macroscopic tumor resection is a strong prognostic factor in patients with EOC, regardless of *BRCA* status [26]. In the current study we did adjust for debulking surgery (yes/no), but the amount of residual disease may be more important in this respect. However, we found no differences between the comparison groups in the percentages of patients with residual disease for those patients with available data. Therefore, we expect no influence on the estimated HRs by adjusting for the amount of residual disease.

In conclusion, in this large case-matched cohort study we confirmed survival benefit for *BRCA1/2*-associated EOC patients treated with mainly platinum-based chemotherapy. This may indicate higher sensitivity to chemotherapy, both in the first-line and in the recurrent setting. The observed benefit appears to be limited to a relatively short period after EOC diagnosis. Future research is warranted to assess in more detail the added value of PARPi on both PFS and OS, especially on the long-term, where the benefit of classic systemic treatment seems to diminish and even disappear.

## Supporting information

**S1 Table. Patient and tumor characteristics–dataset for prospective analyses.**
(DOCX)

**S2 Table. Association of *BRCA1* and *BRCA2* germline pathogenic variant status with progression-free survival and overall survival for the left-truncated analyses.**
(DOCX)

## Acknowledgments

The Hereditary Breast and Ovarian Cancer Research Group Netherlands (HEBON) consists of the following Collaborating Centers: Netherlands Cancer Institute (coordinating center), Amsterdam, NL: M.A. Rookus, F.B.L. Hogervorst, F.E. van Leeuwen, M.A. Adank, M.K. Schmidt, D.J. Stommel-Jenner, R. de Groot; Erasmus Medical Center, Rotterdam, NL: J.M. Collée, A.M.W. van den Ouweland, M.J. Hooning, I.A. Boere; Leiden University Medical Center, NL: C.J. van Asperen, P. Devilee, R.B. van der Luijt, T.C.T.E.F. van Cronenburg; Radboud University Nijmegen Medical Center, NL: M.R. Wevers, A.R. Mensenkamp; University Medical Center Utrecht, NL: M.G.E.M. Ausems, M.J. Koudijs; Amsterdam UMC, University of Amsterdam, NL: I. van de Beek; Amsterdam UMC, Vrije Universiteit Amsterdam, NL: K. van Engelen, J.J.P. Gille; Maastricht University Medical Center, NL: E.B. Gómez García, M.J. Blok, M. de Boer; University of Groningen, NL: L.P.V. Berger, A.H. van der Hout, M.J.E. Mourits, G.H. de Bock; The Netherlands Comprehensive Cancer Organization (IKNL): S. Siesling, J. Verloop; The nationwide network and registry of histo- and cytopathology in The Netherlands (PALGA): E.C. van den Broek. HEBON thanks the study participants and the registration teams of IKNL and PALGA for part of the data collection. The authors thank Dr. M. Kriege and Dr. C. Seynaeve for their intellectual contribution to the design of the study.

## Author Contributions

**Conceptualization:** Bernadette A. M. Heemskerk-Gerritsen, Antoinette Hollestelle, Maartje J. Hooning, Ingrid A. Boere.

**Data curation:** Bernadette A. M. Heemskerk-Gerritsen.

**Formal analysis:** Bernadette A. M. Heemskerk-Gerritsen.

**Investigation:** Bernadette A. M. Heemskerk-Gerritsen.

**Methodology:** Bernadette A. M. Heemskerk-Gerritsen.

**Resources:** Christi J. van Asperen, Klaartje van Engelen, Encarna B. Gómez Garcia, Joanne A. de Hullu, Marian J. E. Mourits, Maartje J. Hooning.

**Supervision:** Maartje J. Hooning, Ingrid A. Boere.

**Visualization:** Bernadette A. M. Heemskerk-Gerritsen.

**Writing – original draft:** Bernadette A. M. Heemskerk-Gerritsen.

**Writing – review & editing:** Antoinette Hollestelle, Christi J. van Asperen, Irma van den Beek, Willemien J. van Driel, Klaartje van Engelen, Encarna B. Gómez Garcia, Joanne A. de Hullu, Marco J. Koudijs, Marian J. E. Mourits, Maartje J. Hooning, Ingrid A. Boere.

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
