## [Decision Letter · Decision Letter 0]

18 May 2022

PONE-D-22-08760Progression-free survival and overall survival after BRCA1/2-associated epithelial ovarian cancer: a matched cohort studyPLOS ONE

Dear Dr. Heemskerk-Gerritsen,

Thank you for submitting your manuscript to PLOS ONE. After careful consideration, we feel that it has merit but does not fully meet PLOS ONE’s publication criteria as it currently stands. Therefore, we invite you to submit a revised version of the manuscript that addresses the points raised during the review process.

We look forward to receiving your revised manuscript.

Kind regards,

Mohammad R. Akbari

Academic Editor

PLOS ONE

Journal Requirements:

Additional Editor Comments:

The two reviewers nicely outlined the issues that need to be addressed in your manuscript before publishing it. Please do your best in addressing their comments and resubmit your revised manuscript for further consideration.

Reviewers' comments:

Reviewer's Responses to Questions

**Comments to the Author**

1. Is the manuscript technically sound, and do the data support the conclusions?

Reviewer #1: Yes

Reviewer #2: Yes

2. Has the statistical analysis been performed appropriately and rigorously? 

Reviewer #1: Yes

Reviewer #2: Yes

3. Have the authors made all data underlying the findings in their manuscript fully available?

Reviewer #1: Yes

Reviewer #2: Yes

4. Is the manuscript presented in an intelligible fashion and written in standard English?

Reviewer #1: Yes

Reviewer #2: Yes

5. Review Comments to the Author

Reviewer #1: Thank you for the opportunity to review this very important analysis aiming to compare survival trajectory of women with ovarian cancer; with and without a germline BRCA mutation. The manuscript is well-written, the methodology is robust, and the analysis is of interest. However, there are a few factors that should be considered that the authors did not describe in detail, address, or account for. Considering the access to the detailed information for such an analysis is available to these authors, I suggest some of the key clinical and treatment variables are included to allow for a more robust analysis. In particular, route and timing of chemo and residual disease status following surgery. Some specific comments include:

1. Introduction: line 69: age-specific recommendations for preventive surgery is based on ages when risks start to increase; not really when childbearing is complete

2. Introduction: line 76: the authors did not include one of the largest, comprehensive analyses on the topic (PMID: 26556769).

3. Methods: line 106: the authors should consider refining the staging as a matching variable. Why were patients not matched by exact stage?

4. Methods; line 108: although clearly addressed in the limitations section of the discussion; this is a major limitation of this study. The sporadic cases were assumed to be negative for BRCA [or other HRD mutations]; which definitely resulted in misclassification of a proportion of these women who were included as the ‘control’ group.

5. Methods: line 134: ‘complete debulking surgery – yes/no’ is not an ideal classification of this very important predictor of prognosis. Ideally, the authors would have had classified this information according to size of residual disease following cytoreductive surgery … unless this indeed means NO or ZERO residual disease.

6. Methods: line 176 – the fact that there was a lot of missing data for the key prognostic variables is also problematic, in particular, residual disease.

7. Results: Table 1 should include overall survival from diagnosis to death

8. Results: Table 1 there is no information on route of chemotherapy; adjuvant vs. neoadjuvant, etc.

9. Results: Table 1 has no information on stage, histology, etc.

10. Results: although very few women used PARPs, perhaps a sensitivity analysis excluding these patients and running the key models would be a good idea.

11. Results: the inclusion of all stages and subtypes is a key limitation. Analyses should be stratified, at the very least, by stage ¾ OR high grade serous disease.

12. Discussion: the discussion is well done and the limitations are well described. Nevertheless, given there are other publications on the topic that allowed for a more robust statistical analysis on the topic.

Reviewer #2: Thank you for this important and well written paper. Though I am no expert on statistics, I thought your analysis was incredibly well thought out and thorough. The paper reads neatly and is easy to understand.

My critical feedback is as follows:

1. You mention that none of the sporadic OC specimens were tested for a somatic mutation. I feel the lack of somatic testing will include a number of BRCA positive specimens in the sporadic group, and will confound your results. I would like to see some attention paid to this in the confounders and discussion, as I think it is a major limitation of your study. I also think you should mention that these are germline mutations only in your abstract and introduction, as readers may now be accustomed to having EOC specimens tested and may assume all specimens are correctly classified before reading your statement on gremlin vs somatic testing.

2. The range of follow up includes 0.1 years. Did you consider a minimum amount of follow up time to include in the criteria (0.5y for example) to exclude the patients who died so early in their journey? At minimum did you match for death/recurrence at less than 6 mo as these outcomes are likely not in the spirit of your conclusions, which is to compare longer term outcomes?

3. Could you expand on censoring events in line 128? I am unclear what you mean by “another tumour” and wonder what you include here. Are metastasis included? Benign tumours?

4. Line 135 has a typographic error and should read “not confounding”

Thank you again for this excellent work. I look forward to its final acceptance.

6. PLOS authors have the option to publish the peer review history of their article (what does this mean?). If published, this will include your full peer review and any attached files.

Reviewer #1: No

Reviewer #2: No

---

## [Author Response · Author response to Decision Letter 0]

11 Jul 2022

Journal Requirements:

Response

We have checked the manuscript and adjusted if necessary.

Response

We have included our full ethics statement in the ‘Methods’ section of the manuscript file (page 4; lines 96-97. Please note that pages and line numbers refer to the Revised Manuscript with Track Changes).

Response

We have compared the ‘Funding Information’ section in the manuscript with the ‘Financial Disclosure’ section and the given information is correct.

4. We note that you have indicated that data from this study are available upon request. PLOS only allows data to be available upon request if there are legal or ethical restrictions on sharing data publicly. For information on unacceptable data access restrictions, please see http://journals.plos.org/plosone/s/data-availability#locunacceptable-data-access-restrictions.

Response

We have addressed the requested prompt in the revised cover letter.

Additional Editor Comments:

The two reviewers nicely outlined the issues that need to be addressed in your manuscript before publishing it. Please do your best in addressing their comments and resubmit your revised manuscript for further consideration.

Response

We would like to thank both reviewers for their time and thorough and fair review of our manuscript. 

Reviewer #1: 

Thank you for the opportunity to review this very important analysis aiming to compare survival trajectory of women with ovarian cancer; with and without a germline BRCA mutation. The manuscript is well written, the methodology is robust, and the analysis is of interest. However, there are a few factors that should be considered that the authors did not describe in detail, address, or account for. 

Considering the access to the detailed information for such an analysis is available to these authors, I suggest some of the key clinical and treatment variables are included to allow for a more robust analysis. In particular, route and timing of chemo and residual disease status following surgery. 

Some specific comments include:

1. Introduction: line 69: age-specific recommendations for preventive surgery is based on ages when risks start to increase; not really when childbearing is complete

Response 

We definitely agree with the reviewer that risk-reducing salpingo-oophorectomy is advised from specific ages, and not based on completeness of childbearing. We have rephrased the sentence on this topic in the Introduction (page 3; lines 69-72. Please note that pages and line numbers refer to the Revised Manuscript with Track Changes). 

2. Introduction: line 76: the authors did not include one of the largest, comprehensive analyses on the topic (PMID: 26556769).

Response

According to the reviewer’s suggestion, we have included the study by Kotsopoulos et al. (reference number 29). 

3. Methods: line 106: the authors should consider refining the staging as a matching variable. Why were

patients not matched by exact stage? 

Response

Since BRCA1/2-associated ovarian cancer is usually diagnosed at considerable younger age than sporadic ovarian cancer, selecting a sufficient number of sporadic controls was difficult when matching on exact stage. Therefore, in consultation with the medical oncologist and in view of the different treatment indications for patients with lower and higher FIGO stages, we matched on the more rough categories of low (i.e. <= IIA) and high (i.e. >=IIB) FIGO stage. Reassuringly, as can be seen in the table below, there were no import differences between the comparison groups when looking at the exact FIGO stage, neither within the low grade categories nor in the high grade categories.

 BRCA1 sporadic BRCA2 sporadic 

 N (%) N (%) p-value N (%) N (%) p-value

FIGO Low (≤IIA) 

I 0 (0%) 5 (11%) 0.277 0 (0%) 2 (11%) 0.383

IA 7 (20%) 10 (22%) 5 (32%) 7 (36%) 

IB 3 (9%) 3 (6%) 2 (12%) 0 (0%) 

IC 18 (53%) 24 (52%) 7 (44%) 8 (42%) 

IIA 6 (18%) 4 (9%) 2 (12%) 2 (11%) 

FIGO High (≥IIB) 

IIB 20 (6%) 25(8%) 0.053 5 (5%) 5 (6%) 0.086

IIC 25 (8%) 26 (8%) 3 (3%) 8 (9%) 

III 9 (3%) 6 (2%) 6 (6%) 1 (1%) 

IIIA 20 (6%) 15 (5%) 5 (5%) 2 (2%) 

IIIB 45 (14%) 39 (13%) 12 (13%) 16 (18%) 

IIIC 145 (45%) 168 (54%) 49 (51%) 51 (57%) 

IV 59 (18%) 31 (10%) 16 (17%) 7 (8%) 

Unknown 32 33 11 14 

For the sake of readability, we omitted these numbers from the table for now, but when desired, we can add them either in the manuscript or as supplementary data.

4. Methods; line 108: although clearly addressed in the limitations section of the discussion; this is a major limitation of this study. The sporadic cases were assumed to be negative for BRCA [or other HRD mutations]; which definitely resulted in misclassification of a proportion of these women who were included as the ‘control’ group.

Response

We totally agree with the reviewer that the fact that not all sporadic patients were DNA tested is a limitation of the study. In addition to the considerations already addressed in the discussion (page 18; lines 286-294), we might speculate that with the same year of diagnosis and the same age at diagnosis – following from the matching criteria – the indication for DNA testing may have been the same for both groups. This may indicate that for sporadic patients with an indication for testing, indeed no BRCA mutation was found. Furthermore, the sporadic patients were in any case not included in the national Hereditary Breast and Ovarian Cancer Netherlands (HEBON) database, which may indicate at least a lack of family history of breast cancer or ovarian cancer. Still, these arguments do not completely rule out potential misclassification. Unfortunately, due to privacy laws and regulations we cannot retrieve and test tumor tissue of those patients who were not tested before.

5. Methods: line 134: ‘complete debulking surgery – yes/no’ is not an ideal classification of this very important predictor of prognosis. Ideally, the authors would have had classified this information according to size of residual disease following cytoreductive surgery … unless this indeed means NO or ZERO residual disease.

Response

‘Complete debulking surgery – yes/no’ indeed means the absence (‘yes’) or presence (‘no’) of any residual disease. For clarification, we have added this specification into the manuscript (page 6; line 136).

6. Methods: line 176 – the fact that there was a lot of missing data for the key prognostic variables is also problematic, in particular, residual disease.

Response

We totally agree with the reviewer that missing data on residual disease is a limitation of the study. However, as mentioned in the Discussion (page 19; lines 312-315), and as can be found in Table1 , we observed no differences between the comparison groups in the percentages of patients with residual disease for those patients with available data. As there seems no association for this variable with the exposure (i.e. carrying a germline BRCA pathogenic variant) in the current study population, the variable is no confounder according to the classical definition for confounders (i.e. being associated with the exposure and the outcome without being an intermediate factor), and therefore the variable does not have to be included in the multivariable model. In addition, although we realize we cannot be absolutely sure of this, we cannot think of a plausible reason to expect this distribution would be different among the patients with missing data, especially since the percentage of these patients is approximately the same in all groups (~30%). Therefore, we expect no influence on the estimated HRs by adjusting for the presence or absence of residual disease 

7. Results: Table 1 should include overall survival from diagnosis to death

Response

We thank the reviewer for this suggestion and added the data into Table 1 and into Supplementary Table S1.

8. Results: Table 1 there is no information on route of chemotherapy; adjuvant vs. neoadjuvant, etc.

Response

We thank the reviewer for this suggestion and added the data into Table 1 and into Supplementary Table S1.

9. Results: Table 1 has no information on stage, histology, etc.

Response

Information on stage can be found above in our response on comment 3. As mentioned there, we omitted these numbers from the table for the sake of readability, but when desired, we can add them in. Following the reviewer’s suggestion, we have added the information on histology into Table 1 and into Supplementary Table S1. 

10. Results: although very few women used PARPs, perhaps a sensitivity analysis excluding these patients and running the key models would be a good idea.

Response

Indeed, only a few women in this cohort were treated with PARP inhibitors (PARPi), and if so this was only done in the recurrent setting. Therefore, there is in our opinion no rationale to exclude patients who received PARPi after recurrent disease from the PFS analyses. Following the reviewer’s suggestion, we performed subgroup analyses for OS excluding patients who were treated with PARPi in the recurrent setting and their matched counterparts. The results can be found in the table below. The subgroup analyses for OS revealed similar results as the original analyses. We have added a few sentences regarding the subgroup analyses for OS to the Discussion (page 17; lines 261-265).

11. Results: the inclusion of all stages and subtypes is a key limitation. Analyses should be stratified, at the very least, by stage 3-4 OR high grade serous disease.

Response

In our opinion, the inclusion of all stages and subtypes is no limitation but rather a reflection of reality. In addition, since the distribution of low and high FIGO stages is not different between the comparison groups (due to the matching procedure), there is no need to adjust for this variable in the multivariable analyses. To inform the reviewer, we performed subgroup analyses including only those patients and their matched counterparts with FIGO stage 3 or 4. The results can be found in the table below. The proportional hazard assumption was violated only for the overall survival analysis comparing BRCA1 gPV carriers with sporadic patients. For the other analyses, stratification by time was not necessary and survival was in favor of the BRCA gPV carriers during the whole observation period. As one can see, the results point in the same direction as the original analyses, with slightly stronger hazard ratios.

 HR (95% CI)

 BRCA1 versus sporadic BRCA2 versus sporadic

Progression-free survival 0.75 (0.61-0.92) 0.46 (0.31-0.67)

Overall survival 0.65 (0.53-0.81) 0.42 (0.28-0.62)

Observation period < 5.1 years 0.56 (0.44-0.72) 

Observation >= 5.1 years 0.98 (0.64-1.49) 

12. Discussion: the discussion is well done and the limitations are well described. Nevertheless, given there are other publications on the topic that allowed for a more robust statistical analysis on the topic.

Response

We thank the reviewer for the kind words regarding the discussion. Indeed, there are other publications on the topic, although only a few have stratified the analyses by observation time or performed separate BRCA1 and BRCA2 analyses.

Reviewer #2: 

Thank you for this important and well written paper. Though I am no expert on statistics, I thought your analysis was incredibly well thought out and thorough. The paper reads neatly and is easy to understand.

My critical feedback is as follows:

1. You mention that none of the sporadic OC specimens were tested for a somatic mutation. I feel the lack of somatic testing will include a number of BRCA positive specimens in the sporadic group, and will confound your results. I would like to see some attention paid to this in the confounders and discussion, as I think it is a major limitation of your study. I also think you should mention that these are germline mutations only in your abstract and introduction, as readers may now be accustomed to having EOC specimens tested and may assume all specimens are correctly classified before reading your statement on germline vs somatic testing.

Response

We agree with the reviewer that due to the lack of somatic testing the sporadic group may contain a number of BRCA positive specimens. Indeed, this may have influenced the results, although we think this influence will be limited, as about only 5% of EOCs have a somatic BRCA pathogenic variant. In addition, under the assumption that survival benefit will also apply to EOCs with a somatic BRCA pathogenic variant, potential misclassification of these EOCs in the sporadic group would led to an underestimation of the observed survival benefit rather than an overestimation. Therefore, although we agree that the lack of data on somatic testing may be a deficiency in the study, in our opinion this may play a minor role. We have added a paragraph in the Discussion on this topic (pages 18-19; lines 298-305).

To clarify more that the BRCA-deficient groups contain only patients with a germline pathogenic variant, we have mentioned this more specific in the abstract and the manuscript (page 2; lines 36, 41 & 55. page 4; line 87. Page 14; line 220. Page 15; line 225). Furthermore, to make this even more clear, we have replaced the abbreviation PV with gPV throughout the manuscript.

2. The range of follow up includes 0.1 years. Did you consider a minimum amount of follow up time to include in the criteria (0.5y for example) to exclude the patients who died so early in their journey? At minimum did you match for death/recurrence at less than 6 mo as these outcomes are likely not in the spirit of your conclusions, which is to compare longer term outcomes?

Response

As previous studies reported different short-term and long-term survival rates for gPV carriers, we were highly interested whether this also applied for our cohort. For this reason, we deliberately included patients who died early after diagnosis. Exclusion or matching these patients would have biased the survival rates for the short-term analyses.

3. Could you expand on censoring events in line 128? I am unclear what you mean by “another tumour” and wonder what you include here. Are metastasis included? Benign tumours?

Response

We considered only malignant tumors as another tumor. The date of the first recurrence defines the endpoint of the PFS analysis, and therefore metastases are not considered as censoring events nor as another tumor. To clarify the definition of the censoring events, we rephrased ‘another tumor’ into ‘a new primary malignant tumor’ (page 6; line 131).

4. Line 135 has a typographic error and should read “not confounding”

Response

We have corrected the typographic error (page 6; line 138).

---

## [Decision Letter · Decision Letter 1]

9 Sep 2022

Progression-free survival and overall survival after BRCA1/2-associated epithelial ovarian cancer: a matched cohort study

PONE-D-22-08760R1

Dear Dr. Heemskerk-Gerritsen,

We’re pleased to inform you that your manuscript has been judged scientifically suitable for publication and will be formally accepted for publication once it meets all outstanding technical requirements.

Kind regards,

Mohammad R. Akbari

Academic Editor

PLOS ONE

Additional Editor Comments (optional):

Reviewers' comments:

Reviewer's Responses to Questions

**Comments to the Author**

1. If the authors have adequately addressed your comments raised in a previous round of review and you feel that this manuscript is now acceptable for publication, you may indicate that here to bypass the “Comments to the Author” section, enter your conflict of interest statement in the “Confidential to Editor” section, and submit your "Accept" recommendation.

Reviewer #1: All comments have been addressed

2. Is the manuscript technically sound, and do the data support the conclusions?

Reviewer #1: Yes

3. Has the statistical analysis been performed appropriately and rigorously? 

Reviewer #1: Yes

4. Have the authors made all data underlying the findings in their manuscript fully available?

Reviewer #1: Yes

5. Is the manuscript presented in an intelligible fashion and written in standard English?

Reviewer #1: (No Response)

6. Review Comments to the Author

Reviewer #1: (No Response)

7. PLOS authors have the option to publish the peer review history of their article (what does this mean?). If published, this will include your full peer review and any attached files.

Reviewer #1: No

---

## [Editor Report · Acceptance letter]

13 Sep 2022

PONE-D-22-08760R1 

Progression-free survival and overall survival after *BRCA1/2*-associated epithelial ovarian cancer: a matched cohort study 

Dear Dr. Heemskerk-Gerritsen:

I'm pleased to inform you that your manuscript has been deemed suitable for publication in PLOS ONE. Congratulations! Your manuscript is now with our production department. 

Kind regards, 

on behalf of

Dr. Mohammad R. Akbari 

Academic Editor

PLOS ONE